# Reappraisal of the Lymphatic Drainage System of the Distal Rectum: Functional Lymphatic Flow into the Presacral Space and Its Clinical Implication in Rectal Cancer Treatment

**DOI:** 10.3390/biomedicines11020274

**Published:** 2023-01-19

**Authors:** Ri-Na Yoo, Hyeon-Min Cho, Bong-Hyeon Kye, Yoon-Suk Lee, Yi-Suk Kim

**Affiliations:** 1Division of Colorectal Surgery, Department of Surgery, St. Vincent’s Hospital, College of Medicine, The Catholic University of Korea, Suwon 16247, Republic of Korea; 2Department of Surgery, Seoul St. Mary’s Hospital, The Catholic University of Korea, Seoul 06591, Republic of Korea; 3Catholic Institute for Applied Anatomy, Department of Anatomy, College of Medicine, The Catholic University of Korea, Seoul 06591, Republic of Korea

**Keywords:** rectal neoplasms, lymphatic vessels, metastasis, fluorescent dyes

## Abstract

Understanding the source and route of pelvic metastasis is essential to developing an optimal strategy for controlling local and systemic diseases of rectal cancer. This study aims to delineate the distribution of lymphatic channels and flow from the distal rectum. In fresh-frozen cadaveric hemipelvis specimens, the ligamentous attachment of the distal rectum to the pelvic floor muscles and the presacral fascia were evaluated. Using indocyanine green (ICG) fluorescence imaging, we simultaneously evaluated the gross anatomy of the lymphatic communication of the distal rectum. We also investigated the lymphatic flow in the pelvic cavity intraoperatively in rectal cancer patients who underwent radical rectal resection with total mesorectal excision (TME). In fresh cadavers, multiple small perforating lymphovascular branches exist in the retrorectal space, posteriorly connecting the mesorectum to the presacral fascia. The lymphatic flow from the distal rectum drains directly into the presacral space through the branches. In patients who underwent TME for rectal cancer, intraoperative ICG fluorescence signals were seen in the pelvic sidewalls and the presacral space. This anatomical study demonstrated that the lymphatic flow from the distal rectum runs directly to the pelvic lateral sidewalls and the presacral space, suggesting a possible route of metastasis in distal rectal cancer.

## 1. Introduction

Lymph node metastasis in rectal cancer is a critical prognostic factor that affects survival outcomes. The pattern of nodal metastasis is related to lymphatic flow from the rectum to different regions in the pelvic cavity. Based on previous anatomic studies of lymphatic flow from the rectum, it is believed that there are three different routes of lymphatic drainage depending on the location within the rectum: the upper route along the superior rectal vessels to the inferior mesenteric artery, the lateral route to the middle rectal vessels to the internal iliac and obturator basin, and the downward route extending to the inguinal lymph nodes [1].

Applying the concept of oncologic surgery, Heald suggested precise dissection preserving the mesorectal fascia, called total mesorectal excision (TME), for rectal mobilization to eliminate possible metastatic lymph nodes along the upper route [2]. TME has become the gold standard in rectal cancer surgery, reducing local recurrence and improving survival outcomes worldwide [3,4]. Adopting a multimodal treatment strategy with neoadjuvant chemoradiotherapy (nCRT) combined with TME could lower the local recurrence rate from 5% to 10% [5]. However, previous reports demonstrated that extramesorectal lymph node metastasis occurs in approximately 10 to 25% of patients who undergo complete TME and achieve a negative circumferential margin [6,7]. Thus, some surgeons insist on lateral lymph node dissection (LLND) because lateral lymph node metastasis through the lateral route is not cleared by TME alone, particularly in locally advanced low rectal cancer [8,9].

The controversy regarding performing lateral pelvic lymph node dissection is still ongoing. Proponents assert that nCRT followed by TME cannot eradicate metastatic lymph nodes in the pelvic sidewalls, and additional lateral lymph node dissection is necessary to gain optimal local control [10,11]. Opponents insist that the rate of metastatic lateral lymph nodes is relatively low, but the rate of postoperative morbidity after LLND is high, thus exceeding the oncologic benefits [12]. Furthermore, the five-year survival rate of patients with lateral pelvic metastasis is approximately 40%, even after LLND [13], and systemic metastasis, usually to the liver or lung, is often found at the time of diagnosis [14]. From this perspective, some clinicians and surgeons consider lateral pelvic lymph node metastasis to be distant metastasis [15]. In addition, the pattern of local recurrence reveals that the recurrence site is not limited to the lateral pelvic sidewalls. More than half of local recurrences occur in the pelvic cavity’s central area, consisting of the presacral space, pelvic floor, and anastomotic areas [16]. When recurrence occurs after complete TME is performed, the source of recurrence is obscure in cases where a negative circumferential margin is achieved, and no lymph node metastasis is found in the mesorectum.

To resolve such controversies and develop an optimal strategy for obtaining local and systemic disease control, the source and route of systemic and locoregional metastasis in distal rectal cancer must be understood. Hypothesizing the existence of lymphatic circulation other than in the lateral pelvic side walls, we aim to re-evaluate the distribution of lymphatic channels and functional flow from the distal rectum to identify the route of metastasis in this study. Using fluorescent imaging, we evaluated the gross anatomy of the lymphatic communication of the distal rectum. We also intraoperatively investigated the lymphatic flow from the distal rectum in the pelvic cavity of rectal cancer patients with the same fluorescent imaging technique.

## 2. Materials and Methods

### 2.1. ICG Fluorescence Imaging of Dissected Cadavers

Three fresh-frozen cadaveric hemipelvis specimens (average age, 81.7 years; range, 70.8–87.8 years; 3 males) were used in this study. The procured hemipelvis specimens included the sacrum, ischium, ilium, pubis to the proximal femur, and related soft tissues. All specimens were confirmed to have no prior surgeries to the abdomen, pelvis, and hip joint, and each was confirmed to have no surgical scars.

The cadavers were hemisected in the midsagittal plane to identify the detailed anatomic structures and facilitate dissection. Before beginning the dissection, 0.2 mL ICG solution, 25 mg dissolved in 10 mL distilled water (Dongindang, Korea), was injected into the submucosa 1 cm above the dentate line. To manually induce lymphatic flow, a massage technique was applied for 20 min on the injection site after the injection [17]. Then, the rectum was mobilized with the standard TME technique, starting from the sacral promontory down to the pelvic floor. The neurovascular bundle and fascial attachment to the rectum were preserved during mobilization. Fluorescence images reflecting the lymphatic flow were obtained using a near-infrared ray (NIR) camera system (TH102 ICG laparoscopy system; Karl Storz).

### 2.2. Intraoperative ICG Fluorescence Imaging

ICG was injected into the submucosal layer at the beginning of the surgery as a routine procedure to distinguish the rectal wall from the pelvic floor fascia during the TME procedure, as shown in Figure 1. A total of 1 mL of the ICG solution prepared as described above was injected into the submucosa of the anterior and posterior rectal wall 1 cm above the dentate line at the beginning of the surgery. After completing the TME and the rectal transection, the ICG-fluorescent signal was examined using the same laparoscopic NIR camera system to detect the lymphatic flow in the pelvis. The institutional review board of St. Vincent’s Hospital, the Catholic University of Korea, approved this study (IRB number VC21RISI0072). The requirement for informed consent was waived by the IRB because of the retrospective nature of the study, and the analysis used anonymized clinical data.

## 3. Results

The macroanatomical examination of the lymphatic drainage from the distal rectum in the fresh cadavers revealed that multiple perforating small lymphovascular branches existed in the retrorectal space, connecting the mesorectum to the presacral fascia posteriorly, as shown in Figure 2a. The color image overlaid with ICG fluorescence, shown in Figure 2b, demonstrates that the NIR camera illuminated ICG fluorescence signals in these lymphovascular branches. A pure ICG fluorescence image under NIR light is shown in Figure 2c. In the hemipelvis of the second cadaver shown in Figure 2d, the presacral space showed a stronger ICG fluorescence signal than in the previously dissected cadaver shown in Figure 2e,f. However, the hemipelvis of the third cadaver, shown in Figure 2g, did not show any ICG fluorescence signal in the presacral space, as shown in Figure 2h,i. In one of the three cadavers, the ICG fluorescence signal in the lateral pelvic sidewalls was observed, as shown in Figure 3.

Table 1 demonstrates the demographics and clinicopathologic characteristics of patients in whom the lymphatic flow in the pelvis was examined intraoperatively. Each patient showed different patterns of ICG fluorescence signals. Two patients showed ICG fluorescence signals in both the pelvic sidewalls and the presacral space. One patient showed a fluorescence signal in the left pelvic sidewall. Three patients showed fluorescence signals only in the right pelvic sidewall. The remaining four patients did not show any fluorescence signal. Figure 4 demonstrates the different patterns of the ICG fluorescence signals observed intraoperatively.

## 4. Discussion

This anatomical study investigated the distribution of the lymphatic channels and their functional flow from the distal rectum to the different pelvic regions using an ICG imaging system. The ICG images obtained from the gross anatomical dissection of fresh cadavers reflected that the lymphatic flow from the distal rectum runs into the pelvic lateral sidewalls and the presacral space. This finding was concordant with the intraoperative ICG images taken from rectal cancer patients after TME and rectal transection. In addition, the intraoperative ICG fluorescence images demonstrated illumination on both sides of the pelvic sidewall and the presacral fascia up to the sacral promontory. This finding implies that functional lymphatic flow exists between the distal rectum and the bilateral pelvic sidewalls and the presacral space.

The lateral spread to the left or right pelvic wall can be explained by channels surrounding the middle rectal vessels. As shown in Figure 1, the middle rectal vessels are often detected during TME, as the small vessels directly penetrate the distal rectum. The middle rectal vessels have been the critical anatomical structure related to the lateral route of metastatic spread in distal rectal cancer. In 1950, Blair demonstrated lymphatic draining from the inferior rectal region following nerves from the pelvic part of the rectum [18]. He mentioned that these channels are not a regular route but a potential course for rectal cancer spread, especially when the higher nodes are blocked. Later, using lymphoscintigraphy, Arnaud also showed a similar pattern of lateral lymphatic drainage to the internal iliac nodes in 50% of control participants, suggesting this accessory drainage path as a potential route of lateral spread [19]. The clinical data in this study demonstrated that some patients exhibited ICG fluorescence signals in the bilateral pelvic sidewalls, consistent with previous findings. Hence, lateral spread to the pelvic sidewalls manifests through interindividual variations in functional flow through lymphatic channels.

Lymphatic fluid spreading to the presacral space was an unexpected route of lymphatic flow found in this study. A recent microanatomical study by Sato [20] demonstrated a functional lymphovascular network in the anorectal region connected to the endopelvic fascia covering the pelvic floor muscles. Similar to our study, the authors showed direct ICG fluorescence illuminating the pelvic floor muscles immediately after the ICG injection during surgery. They suggested that the tissue fluid from the distal rectum is readily transported via an extensive lymphovascular network between the distal rectum and its adjacent tissues, partly explaining the high recurrence rate in cases of local excision of early carcinoma in the distal rectum and the worse survival outcomes of advanced rectal cancer requiring extralevator abdominoperineal resection. Furthermore, the cadaveric study by Stelzner [21] showed that the parietal pelvic fascia and the presacral fascia fused at the fourth sacral vertebrae in the midline, densely connected to the posterior rectal wall via the rectosacral ligament. The authors demonstrated that the neurovascular branches were enclosed within the fused fascia layers, and those neurovascular branches were located close to the rectal wall on histologic examination. These studies support our finding of functional lymphatic flow to the presacral space, suggesting a potential route of metastasis.

Although the multidisciplinary approach of neoadjuvant chemoradiotherapy combined with TME has reduced the local recurrence rate to approximately 5% to 10% [22,23], systemic metastasis is still a leading cause of death in rectal cancer patients. Thus, a high rate of systemic metastasis is still problematic [24]. Chen et al. demonstrated that cancer located in the mid to low rectum, a predictive factor for distant metastasis, exhibited a higher hazard ratio to develop distant metastasis than cancers in the upper third rectum [14]. They also showed that patients with low rectal cancer had significantly worse survival outcomes than those with upper rectal cancer: the 5-year overall survival rate was 64% in patients with lower rectal cancer vs. 96% in those with upper rectal cancer [14]. In addition, in an evaluation of the oncologic outcomes of patients who underwent intersphincteric resection after neoadjuvant chemoradiotherapy for low rectal cancer, Park et al. showed that the overall 3-year disease-free survival was 64.9%, and distant metastasis occurred in 22% of patients [25]. These data indicate that cancer in the distal rectum may have distinct anatomical features related to the high rate of distant metastasis. Our anatomical study may provide meaningful data that some individuals may have functionally patent lymphatic flow to the lateral pelvic sidewalls or the presacral space that may dissipate systemic micrometastasis.

Additionally, this partly explains the pattern of local recurrence occurring in the presacral space, pelvic floor, and extraluminal anastomotic areas. Unfortunately, this study did not include a risk factor analysis of patients with functional lymphatic flow. Therefore, further studies elucidating the factors influencing functional lymphatic flow are necessary.

This study had several limitations. First, only a small number of cadavers were used for gross dissection. In addition, intraoperative observation was conducted on a small number of patients. A prospective observational study with a large sample size is necessary to verify our results. Additionally, no microanatomical evaluations were carried out with immunohistochemical staining of the lymphatic channels. Detailed histologic analyses of cadaver and resected specimens of the distal rectum and its adjacent tissue may help delineate the distribution of lymphatic channels. Moreover, the effect of nCRT was not assessed in this study. It is well known that radiotherapy induces an inflammatory response causing structural fibrosis and edema in the radiation field [26]. The change in the anatomic structure of the microenvironment in the patients who underwent nCRT may influence the ICG fluorescence. However, the degree of nCRT effect in each patient is difficult to measure. Further investigation of the microscopic structural change after chemoradiation is necessary to understand the functional lymphatic flow after nCRT.

## 5. Conclusions

This study suggests that functional lymphatic flow exists between the distal rectum, the bilateral pelvic sidewalls, and the presacral space. Furthermore, among patients with distal rectal cancer, certain individuals with functionally patent lymphatic flow may have an increased risk for distant metastasis. Thus, this exploratory investigation may provide noteworthy information regarding treatment strategies, particularly for patients with distal rectal cancer.

## Figures and Tables

**Figure 1 biomedicines-11-00274-f001:**
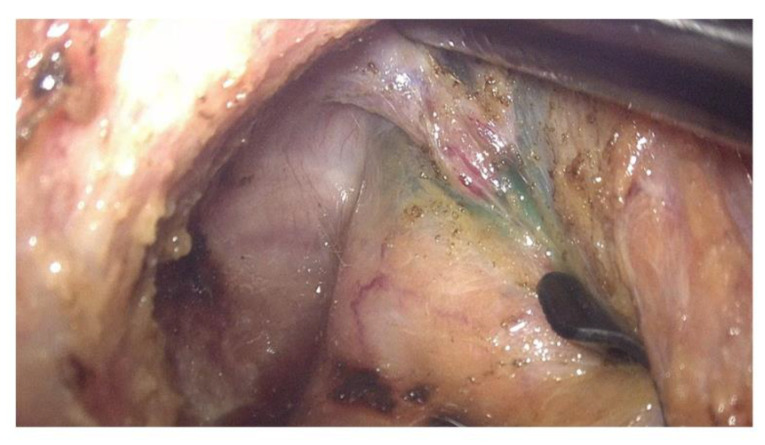
ICG stained rectal wall along with the middle rectal vessels running through the rectum.

**Figure 2 biomedicines-11-00274-f002:**
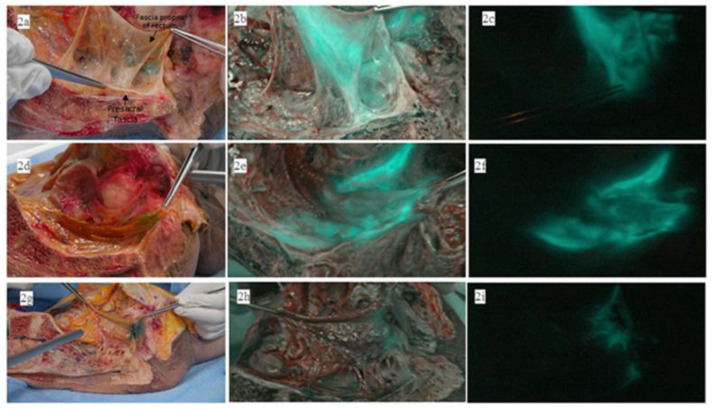
The macroanatomical examination of the lymphatic drainage from the distal rectum in fresh cadavers. (**a**,**d**,**g**) Multiple perforating small lymphovascular branches connect the mesorectum to the presacral fascia; (**b**,**e**,**h**) the color image overlaid with ICG fluorescence; (**c**,**f**,**i**) the pure ICG fluorescence image under NIR light.

**Figure 3 biomedicines-11-00274-f003:**
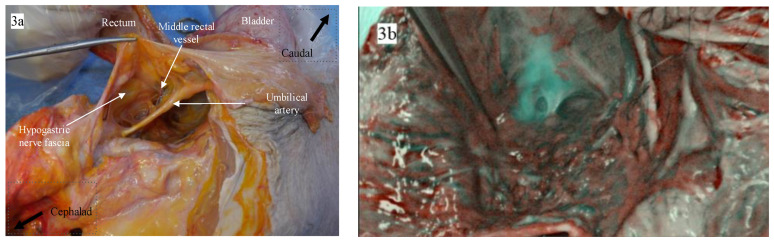
The ICG fluorescence signal in the lateral pelvic sidewalls. (**a**) The gross appearance of the lateral pelvic sidewall in a color image. (**b**) The color image overlaid with the ICG fluorescence. (**c**) The he pure ICG fluorescence image under NIR light.

**Figure 4 biomedicines-11-00274-f004:**
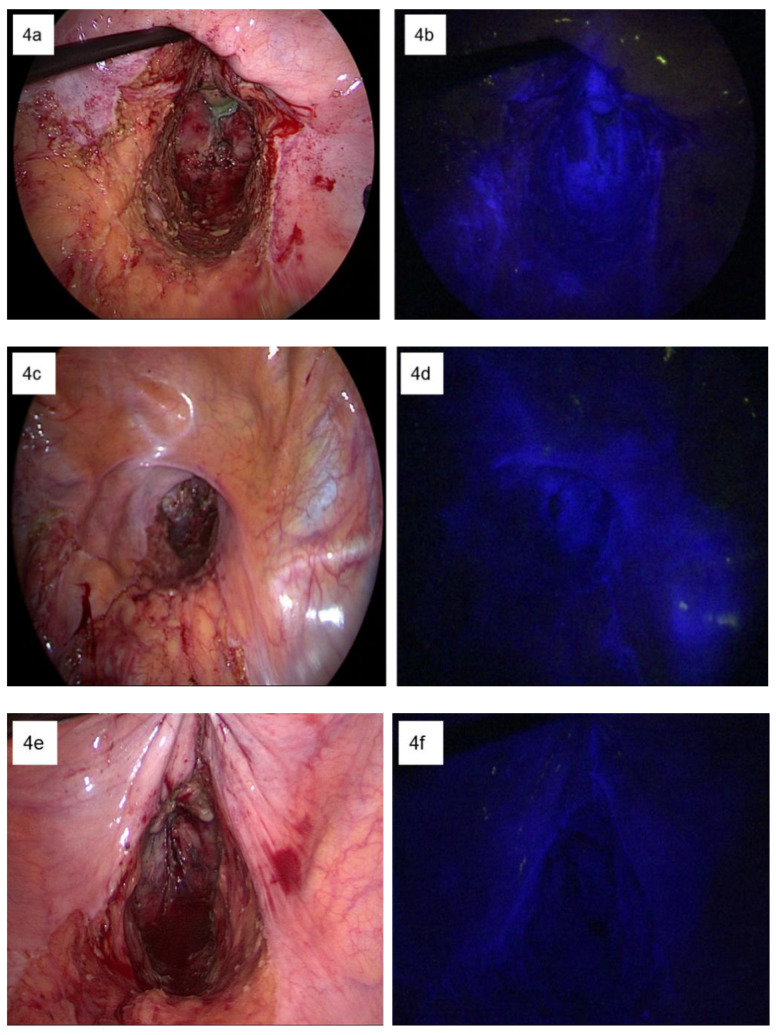
The different patterns of ICG fluorescence signals observed intraoperatively. (**a**,**c**,**e**,**g**,**i**) The gross appearance of the pelvic floor after the rectal transection in color images of different patients. (**b**) The pure ICG fluorescence image under NIR light indicates the bilateral lymphatic flow. (**d**) The pure ICG fluorescence image under NIR light indicates the lymphatic flow only to the right side. (**f**) The pure ICG fluorescence image under NIR light indicates the lymphatic flow only to the left side. (**h**) The pure ICG fluorescence image under NIR light indicates the lymphatic flow to the bilateral pelvic side wall as well as to the presacral area up to the bifurcation of the common iliac vessels. (**j**) The pure ICG fluorescence image under NIR light indicates the lymphatic flow only to the presacral area up to the bifurcation of the common iliac vessels.

**Table 1 biomedicines-11-00274-t001:** Patient demographics, clinicopathology, and ICG fluorescent detection site.

Patient Number	Sex	Age	Tumor The Location from the Anal Verge (cm)	nCRT	Type of nCRT	Operation	Pathologic TNM Stage (p or yp)			ICG Fluorescent Detection Site		
							T	N	M	Right Pelvic Side Wall	Left Pelvic Side Wall	Presacral Space
1	F	69	6	Yes	Long course	Lap LAR	3	0	0	-	-	-
2	M	46	8	Yes	Long-course	Lap LAR	3	1a	0	-	-	-
3	M	60	10	No	-	Lap LAR	3	2a	0	-	-	-
4	M	50	2	Yes	Long-course	Lap ISR	3	1a	0	+	+	+
5	M	61	2	Yes	Long-course	Lap ISR	0	0	0	-	+	-
6	M	68	10	No	-	Lap LAR	4a	2a	0	+	-	-
7	M	66	10	Yes	Long-course	Lap LAR	3	1a	0	+	-	-
8	M	68	10	No	-	Lap LAR	2	0	0	+	-	-
9	M	64	7	Yes	Long-course	Lap LAR	1	0	0	-	-	-
10	F	77	10	No	-	Lap LAR	3	0	0	-	-	+

ICG, indocyanine green; nCRT, neoadjuvant chemoradiotherapy; Lap, laparoscopic; LAR, low anterior resection; ISR, intersphincteric resection; p, pathologic; yp, pathologic after neoadjuvant therapy.

## Data Availability

The authors will share the original data upon request.

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
