# Peer review of "Reappraisal of the Lymphatic Drainage System of the Distal Rectum: Functional Lymphatic Flow into the Presacral Space and Its Clinical Implication in Rectal Cancer Treatment"

_biomedicines, 2023, doi:10.3390/biomedicines11020274_

Round 1

Reviewer 1 Report (Previous Reviewer 1)

The paper is o.k. now for publication.

Author Response

Thank you very much for reviewing the article. All the authors greatly appreciate your effort.

Reviewer 2 Report (New Reviewer)

This is an interesting manuscript that analyzes the distribution of lymphatic channels and their functional flow from the distal rectum to the different pelvic regions using an indocyanine green (ICG) imaging system. The authors hypothesized the existence of lymphatic circulation other than the lateral pelvic side walls and, for this reason, aimed to re-evaluate the distribution of lymphatics and functional flow from the distal rectum. Through cadaveric and intraoperative ICG imaging, they showed that lymphatic flow from the distal rectum runs into the lateral lateral pelvic walls and presacral space. Lymphatic drainage to the presacral space is an unexpected route of lymphatic flow found in this study. For this reason, even if this is only a morphological and anatomical study, it may provide meaningful data on previously unknown routes of systemic metastasization. Moreover, these findings partly explain the pattern of local recurrence occurring in the presacral space, pelvic floor, and extraluminal anastomotic areas. However, this work has several limitations and further studies are needed to elucidate the factors influencing functional lymphatic flow. I just wish the authors would remove the yellow highlighting from some sentences and do a spelling and punctuation check.

Author Response

Thank you very much for reviewing the article. All the authors deeply appreciate your comment. Indeed, the authors sincerely agree on the limitations of this study and the necessity of further research. We are planning on a series of the functional anatomy of the pelvis related to lymphatic flow. We hope our upcoming studies could contribute to elucidating the functional lymphatic flow of the pelvis. Following your comment, we removed the yellow highlights and grammar check. 

Reviewer 3 Report (New Reviewer)

In this study the authors showed that the lymphatic flow from the distal rectum runs directly to  the pelvic lateral sidewalls and the presacral space could be a way of metastasis in distal  rectal cancer.

Major comments

1- Same/ similar findings are known and published previously and  the role of lymphatic flow in CRC is documented. 

PMID: 13182113,  PMID: 35461198PMID: 35257141PMID: 34986999, PMID: 34240110, PMID: 33711144, PMID: 32770742PMID: 32109846 (similar title)

2- In Methodology, the authors mentioned 3 male patients were enrolled in the study . While table 1 include male, female , different categories . It is very confusing.

3- The manuscript is not organized, and confused. I would suggest the authors do a video showing the flow of dye 

4- Figure 4 the quality of  4b,4d,4f,4h,4J are very poor.

5- Table 1 includes missing and confused information.  a) What are number under the column tumor location?  please revise either the title or provide the exact location. b) T,N,M are not mentioned.  Under the sub-column "N"; 1a, what a stands for?

 6- How therapy and cancer stage/location affect the lymphatic flow in CRC metastasis. 

Author Response

Dear Reviewer,

Thank you for your prudent comments. All the authors sincerely appreciate your comments. Your effort immensely helped to improve this article. We earnestly tried to answer your comments and question in the attached file. Thank you again for reviewing this article.   

Round 2

Reviewer 3 Report (New Reviewer)

The authors replied to some of my comments, the videos  are very helpful. In the present form, the manuscript could be publishable. 

This manuscript is a resubmission of an earlier submission. The following is a list of the peer review reports and author responses from that submission.

Round 1

Reviewer 1 Report

The paper is well structured and written. The impact for treatment individualization by surgery/neo RCT may be better outlined e.g. by hypothesis.

The effect of neo-RCT on your findings may be adressed.

Unfortunately Table 1 includes not your ICG findings, demonstrating the individual variations that you found. You may apply 2 tables (1a and 1b)

In your Figures, even for a rectal cancer surgeon it is difficult to see the rectum. May be, it becomes clearer, when you are showing a broader view.? 

The paper is of better quality, if you make the additions/changes suggested. It should be published.

Author Response

Thank you very much for reviewing this article. The authors believe that your comments and suggestions helped improve the article immensely.

As you suggested, the authors added the hypothesis in the introduction, distinguished in highlight. Also, in Table 1, the ICG fluorescence detected in each patient was added.

Regarding the effect of neoadjuvant chemoradiotherapy, the authors think that functional lymphatic flow can be influenced by chemoradiation. We think chemoradiation's therapeutic effect disrupts the lymphatic flow by an inflammatory response. However, the inflammatory response is different in each patient and difficult to measure in this study. The ultimate goal is to understand who is not responding to chemoradiotherapy and exhibits persistent functional flow, with a potential risk factor for locoregional or distant metastasis. We commented on the effect of neoadjuvant chemoradiotherapy as the limitation of the study, highlighted as yellow.

Responding to the comment on the Figure images, we tried to show the ICG fluorescence after the rectal transection in the Figures. You are right about not being able to see the rectum. You can only see the pelvic cavity without the rectum in the gross pictures. In 3D real images, it would be better to distinguish the anatomic structures. Unfortunately, the pictures are only available in 2D images.

Reviewer 2 Report

Manuscript entitled "Reappraisal of the lymphatic drainage system of the distal rectum: functional lymphatic flow into the presacral space and its clinical implication in rectal cancer treatment"

The work is limited by several factors:

1. The case number is too small and should be improved by adding more cases.
2. The translational significance of this work is very limited. There is no clinical advantage identified.

Author Response

Thank you for your comments.

The authors understand that the case number is insufficient to draw a firm conclusion. However, this study attempted to suggest a possible mechanism of local and distant metastasis in distal rectal cancer using cadaver dissection. We are in the process of constructing our database prospectively with clinical data. We hope to conduct a larger cohort study evaluating whether the functional lymphatic flow is related to local or systemic disease.

To understand the high rates of systemic metastasis and local recurrence in patients with distal rectal cancer, the authors hypothesized the existence of functional lymphatic flow in the distal rectum. Using cadaver dissection and ICG fluorescence, we evaluated possible lymphatic channels. We also conducted in vivo tests in patients undergoing radical rectal resection. We found that each patient has a distinctly different pattern of ICG fluorescence, implying interindividual variation in functional lymphatic flow. This finding may influence determining treatment strategy, such as whether to perform lateral pelvic lymph node dissection in cases of positive fluorescent in pelvic side walls. We are still in the early investigation phase, exploring and suggesting the potential mechanism of metastasis in distal rectal cancer.        

Round 2

Reviewer 1 Report

Paper is o.k. now. Next paper on that topic with clinical relevance would be fine. e.g.: Did neoRCT influence this lymphatic distribution of tumor cells effectively or should planning of RT better aim towards this way of lyphatic spread.

Reviewer 2 Report

There is no significant improvement made.